# Evaluation of knowledge and attitudes regarding Alzheimer's disease and related dementia among medical students in Palestine: A cross-sectional study

**Mohammad Abuawad**[1]\*, **Ahmad Rjoub**[2], **Yazan Dumaidi**[3], **Motaz Daraghma**[4], **Mustafa Ghanim**[1], **Maha Rabayaa**[1], **Johnny Amer**[5]

**1** Faculty of Medicine and Health Sciences, Department of Biomedical Sciences, An-Najah National University, Nablus, Palestine, **2** Faculty of Medicine and Health Sciences, Department of Medicine, An-Najah National University, Nablus, Palestine, **3** Intern Medical Doctor, Rafeedia Surgical Hospital, Nablus, Palestine, **4** Faculty of Medicine, Research and Teaching Assistant, Arab American University, Jenin, Palestine, **5** Faculty of Medicine and Health Sciences, Department of Allied and Applied Medical Sciences, An-Najah National University, Nablus, Palestine

\* m.abuawad@najah.edu

## Abstract

### Introduction

Dementia, a major global health concern, is an acquired disorder that causes a progressive decline in cognitive abilities, affecting learning and memory, language, executive function, complex attention, perceptual-motor skills, and social cognition. Our study aims to evaluate the knowledge and attitudes regarding dementia and Alzheimer's disease among medical students.

### Methods

This cross-sectional study was conducted among 393 medical students in Palestine from August 2023 to November 2023. The assessment of knowledge and attitude toward dementia was measured using the Alzheimer's Disease Knowledge Scale (ADKS) and Dementia Attitude Scale (DAS). The data were analyzed using SPSS version 26, and the Mann-Whitney U-test and the Kruskal-Wallis test were used to compare the mean between the groups with a 5% significance level.

### Results

The overall mean score of the student's knowledge of dementia measured by the ADKS was 18.91 (±3.32 SD) out of 30. The mean score of the student's attitude toward dementia measured by the DAS was 91.68 (±3.32 SD). Clinical students had higher ADKS scores than pre-clinical students (p-value < 0.001). No significant differences in the knowledge and attitudes toward dementia were found between males and females. The medical students' knowledge and attitude scores were positively correlated (ρ = 0.227, p-value <0.001).

**Data Availability Statement:** All relevant data are within the manuscript and its Supporting Information files.

**Funding:** The author(s) received no specific funding for this work.

**Competing interests:** The authors have declared that no competing interests exist.

## Conclusion

Palestinian medical student's knowledge about Alzheimer's disease and dementia is insufficient, with students in the clinical phase showing better understanding than pre-clinical students. The findings highlight a necessity for enhancing the dementia curriculum and conducting further studies to evaluate training's impact on students' knowledge and attitudes.

## 1. Introduction

Dementia is an acquired disorder that causes cognitive abilities to decline over time, affecting learning and memory, language, executive function, complex attention, perceptual-motor skills, and social cognition [1,2]. Dementia, a major global health concern, currently affects over 55 million people worldwide [3], with low- and middle-income countries accounting for more than 60% of cases. Each year, nearly 10 million new cases are documented [4]. The varied causes of dementia stem from various diseases and injuries affecting the brain, with Alzheimer's disease being the most common, accounting for 60–70% of cases [5]. According to the World Health Organization, dementia is currently the seventh leading cause of death and is a significant contributor to disability and dependency in the elderly on a global scale [6]. The global economic burden of dementia reached a staggering 1.3 trillion US dollars in 2019 [7].

Dementia is clinically defined as the loss of cognitive functioning (thinking, remembering, learning, and reasoning) and behavioral abilities are lost to the point where they interfere with a person's quality of life and activities. Although memory loss is common, it is not the only sign of dementia [8]. The concept of dementia is still perceived as a normal aging process by patients, caregivers, and even medical practitioners. As per a study in Australia, conducted on adults aged 40–65 years old, it has been noted that 50% of the participants are not able to have a meaningful conversation with patients with dementia [9]. Another study conducted in Qatar demonstrated an average score of 5.3 out of 7 (approximately 75%) when filling the seven measures of knowledge regarding AD and dementia by healthcare professionals showed moderate knowledge of Alzheimer's disease (AD) and dementia. Physicians had the most knowledge, followed by students, educators, researchers, nurses, and others [10].

In the State of Palestine, the prevalence of dementia among individuals aged 50 and older is 2.08%, rising to 4.01% in those aged 60 and older [11]. This highlights the significance of dementia as an essential clinical entity that healthcare practitioners face in their practice. It emphasizes the importance of evidence-based education, which is supported by data on medical education outcomes. According to our review of the literature, such data is lacking, emphasizing the importance of conducting this study. In this study, we aim to evaluate the awareness and attitude of medical students in the State of Palestine regarding Alzheimer's disease and dementia.

## 2. Methods

### 2.1. Setting and study design

The study took place among the Palestinian Medical Students enrolled in the "Medical Doctor" program in the seven medical schools in five provinces in Palestine: An-Najah National University, Al-Azhar University-Gaza, Palestine Polytechnic University, Arab American University, Islamic University of Gaza, Al-Quds University, and Hebron University to gauge the

efficacy of medical education systems in the Palestinian medical faculties in incorporating the Alzheimer's disease (AD) and dementia concepts within the existing medical curricula, a comprehensive assessment is required. The neuroscientific and neuroanatomical courses during the foundational years counting three hours of credit cover AD and dementia in their intended learning objectives. Furthermore, these subjects are addressed during the clinical years within the three-week neurology rotation. The study was done as a cross-sectional online survey undertaken to gauge these parameters of dementia and Alzheimer's disease among medical students in Palestine. The study participants were recruited between 22-08-2023 and 07-11-2023. The study design allowed us to collect data from the different geographical areas of the Palestinian medical faculties in the West Bank and Gaza through an online accessible form that was shared on all the students' platforms. All undergraduate medical students from the medical schools in Palestine were included in the study except the First-year students. The reason behind excluding the first-year students is that this study was carried out during their first semester of Medical School which includes no courses that tackle Alzheimer's disease or dementia. Medical students who did not consent to participate in this study were excluded. A convenient sampling technique was used, and a total of 393 medical students participated in this study. The participants were recruited by sharing the questionnaire using Google form on their study groups, courses, and university emails. The sample size was calculated using the Raosoft ® Sample Size Calculator. The formula for determining the sample size (n) and margin of error (E) is expressed as follows: $n = N^{*} x / ((N-1) E^{2} + x)$, where N represents the total population size. Given a margin of error (E) of 5% and a confidence level of 95%, with a population size (N) of 15000 and a response distribution of 50%, the formula utilizes a z score of 1.96. The minimum recommended sample size was 375. Having an adequate sample size helped reduce the random error.

## 2.2. Ethical and consent approval

Ethical approval was taken from the institutional review board at An-Najah National University (Ref: Med. August. 2023/34). A statement of the study's objectives and purposes was included in the opening section of the questionnaire. The statement included that the collected data would be evaluated and analyzed anonymously and no personal identifiers were collected. The participants were informed of the voluntary nature of their participation in this study and they could withdraw from the study at any time. They were informed that by submitting their response to the questionnaire on Google Forms, they are considered to have agreed to willingly participate in this study. In addition, no personal information was obtained from the participants.

## 2.3. Data collection instrument

The data collection was collected through a survey using Google Forms and evacuated into Google Spreadsheets. The survey was divided into three sections: the first section included sociodemographic information: age, gender, year of study, university, marital status, residence type, experience with dementia, and source of information. The second section included the 30 items of the Alzheimer's Disease Knowledge Scale (ADKS), a validated and reliable measurement tool [12], while the third section included the Dementia Attitudes Scale (DAS), a scale with established validity and reliability [13]. The ADKS consists of 30 True/False items divided into 7 domains, tackling the symptoms (4 items), risk factors (6 items), assessment and diagnosis (4 items), course of the disease (4 items), life impact (3 items), caregiving (5 items), and treatment (4 items). The incorrect answer was given 0 points while the correct answer was given 1 point, and the sum of the correct answers was used to calculate the total

ADKS. We used the ADKS as it is suitable to assess the knowledge of Alzheimer's disease among medical students [14]. The DAS is a scale consisting of 20 items, with a 7-point Likert scale. The responses to the Likert scale were "Strongly Disagree", "Disagree", "Slightly Disagree", "Neutral", Slightly Agree", "Agree", and "Strongly Agree". The total scores of the responses to the 20 items of the DAS range from 20 to 140. It is divided into two main domains, the Dementia Knowledge domain (items 3,7,10,11,12,14,15,18,19,20), and the Social Comfort domain (items 1,2,4,5,6,8,9,13,16,17). Each of the two domains has scores ranging from 10 to 70, and the sum of the two domains gives the total DAS score. The negatively worded items of the DAS (2, 6, 8, 9, 16, and 17) were reversely coded. The DAS was used to assess the medical students' attitudes toward Alzheimer's disease and related dementia [13]. The DAS was developed to measure attitudes towards dementia for both college students and direct care workers [13]. DAS internal consistency was originally reported using Cronbach's alpha value of 0.83–0.85. The ADKS test-retest reliability coefficient was 0.81 and Internal Consistency Cronbach's alpha was 0.71 [12]. Furthermore, the questionnaire reliability was reevaluated by conducting a pilot study on 30 students and the total Cronbach alpha was 0.77. As a result, the questionnaire's validity and reliability were assured before conducting the study. The questionnaire was in English, as it is the official teaching language in the medical schools in Palestine.

## 2.4. Data analysis

The statistical package of Social Science (SPSS) version 26.0 and Google Spreadsheets were used for analyzing the data in this study. The sociodemographic characteristics, ADKS and its domains, and the DAS and its domains were described using descriptive statistics. Qualitative data were described as frequencies and percentages, while quantitative data were described as means and standard deviations. The Kolmogorov-Smirnov test was used to assess the normality of the continuous data, and it revealed that the ADKS (p-value < 0.001), the scores of all the ADKS domains (p-value < 0.001), the DAS (p-value < 0.001), the scores of the dementia knowledge domain (p-value < 0.001), and the score of the Social Comfort domain (p-value < 0.001) were not normally distributed. Therefore, the Mann-Whitney U-test and the Kruskal-Wallis test were used to compare the means between the groups [15]. The chi-square test was the statistical approach that was used in order to determine the differences between the groups. The Spearman correlation by ranks was used to evaluate the correlation between the ADKA and DAS. Significant differences between groups have been assessed and associations found are highlighted later on in the discussion. The significance level was set at 5%.

## 3. Results

### 3.1. Sociodemographic characteristics of medical students

A total of 393 medical students from all of the medical faculties in Palestine responded to the questionnaire. Notably, 83% of the students were aged 20 years or older, with the remaining 17% of them falling below the age of 20. The mean age for all of the participants was 20.91 ±1.60 years. Gender distribution revealed 53.4% females and 46.6% males. The students from the pre-clinical study phase who participated in our study were about 54.7% of the total participants, while 45.3% were from the clinical study phase. The majority of the students were single (98.5%), and 60.1% of them lived in urban areas. Exploring their familiarity with dementia, only 32.6% of students reported prior experience with dementia, and their major source of information about dementia was the Internet (47.3%), followed by the curriculum (45.6%). The sociodemographic characteristics of the participants are presented in Table 1.

**Table 1. Sociodemographic characteristics of the medical students.**

| Variables | Frequency | Percentage (%) |
|---|---|---|
| **Age:** | | |
| Age < 20 | 67 | 17% |
| Age ≥ 20 | 326 | 83% |
| **Gender:** | | |
| Male | 183 | 46.6% |
| Female | 210 | 53.4% |
| **Medical Student Status:** | | |
| Pre-Clinical: | 215 | 54.7% |
| Clinical | 178 | 45.3% |
| **Marital status:** | | |
| Single | 387 | 98.5% |
| Married/Engaged | 6 | 1.5% |
| **Experience with dementia:** | | |
| Yes | 128 | 32.6% |
| No | 265 | 67.4% |
| **Residence type:** | | |
| City | 236 | 60.1% |
| Village | 136 | 34.6% |
| Camp | 21 | 5.3% |
| **Source of information** | | |
| Internet | 186 | 47.3% |
| Curriculum | 180 | 45.8% |
| Awareness Campaigns | 27 | 6.9% |

## 3.2. Knowledge of medical students toward Alzheimer's disease and dementia

The mean score on the Alzheimer's Disease Knowledge Scale (ADKS) was 18.91 (±3.32 SD) out of 30, representing 63% of the maximum score, with the lowest and highest scores of 10 and 30 respectively. Scores across ADKS domains varied from 45.7% to 78%. Notably, the items of the risk factors domain had the lowest percent of correct scores, while the items of the assessment and diagnosis domain had the highest percent of correct scores. Clinical students had higher ADKS scores than pre-clinical students (U = 14828, Z = 3.861, and p-value < 0.001). There were no significant differences in the scores of the ADKS domains between the pre-clinical and clinical students except for the assessment and diagnosis domain, in which the clinical students had higher scores in the assessment and diagnosis domain than pre-clinical students, with U = 14213, Z = 4.669, and p-value < 0.001. The mean scores of the ADKS domains and the comparison in scores between the pre-clinical and clinical students are presented in Table 2.

Table 3 presents the 30 statements of the ADKS divided into the 7 domains, along with the corresponding correct answers. The most correctly answered item from the ADKS was "Alzheimer's disease is one type of dementia." from the assessment and diagnosis category, in which about 88% of the students answered the correct answer. The least correctly answered item from the ADKS was "When people with Alzheimer's disease begin to have difficulty taking care of themselves, caregivers should take over right away." from the caregiving category in which only 18% of the students answered the correct answer.

**Table 2. The correct answers of content domains on the Alzheimer's disease knowledge scale (n = 393).**

| Domain | Range of total Scores | Mean ± SD | Mean correct rate (%) | Pre-Clinical, Mean Ranks | Clinical, Mean Ranks | U | Z | p-value[a] |
|---|---|---|---|---|---|---|---|---|
| Symptoms | 0–4 | 2.74 ±0.94 | 68.5% | 190.48 | 204.87 | 17734 | 1.311 | 0.190 |
| Risk Factors | 0–6 | 2.74 ±1.28 | 45.7% | 193.97 | 200.65 | 18484.5 | 0.596 | 0.551 |
| Assessment & Diagnosis | 0–4 | 3.12 ±0.87 | 78% | 174.11 | 224.65 | 14213 | 4.669 | <0.001* |
| Course | 0–4 | 2.82 ±0.98 | 70.5% | 189.15 | 206.48 | 17447.5 | 1.588 | 0.112 |
| Life Impact | 0–3 | 2.03 ±0.71 | 67.7% | 188.88 | 206.81 | 17289.5 | 1.743 | 0.081 |
| Caregiving | 0–5 | 2.63 ±1.05 | 52.6% | 189.34 | 206.26 | 17487.5 | 1.539 | 0.124 |
| Treatment | 0–4 | 2.83 ±0.89 | 70.7% | 189.08 | 206.56 | 17432.5 | 1.616 | 0.106 |
| **Total ADKS** | 0–30 | 18.91±3.23 | 63% | 176.97 | 221.19 | 14828 | 3.861 | <0.001* |

SD: Standard deviation

* Significant at P value <0.05.

[a] Mann-Whitney U-test.

### 3.3. Attitudes of medical students toward dementia

The mean Dementia Attitude Scale (DAS) was 91.68 (±3.32 SD) out of 140, representing 65.5% of the maximum score, with the lowest and highest scores of 54 and 126, respectively. The mean social comfort score was 44.27 (±7.61 SD) out of 70, with a 63.2% mean correct rate. The mean dementia knowledge score was 47.41 (±13.04 SD) out of 70, with a 67.7% mean correct rate. No statistically significant differences were observed in the DAS scores, Social comfort scores, or dementia knowledge Scores between the pre-clinical and clinical students. Detailed mean scores of the DAS domains and the comparative analysis between the pre-clinical and clinical students are presented in Table 4.

The results obtained for DAS items are presented in Table 5. About 81.4% of the students disagree with the statement that they feel afraid of people with ADRD, and 70% of them disagree with the statement that they feel uncomfortable being around people with ADRD. Moreover, 55.7% of the students disagree that they cannot imagine taking care of someone with ADRD. In the knowledge domain, about 71.5% of the students agree that every person with ADRD has different needs, while 71.8% of them agree that it is important to know the past history of people with ADRD. Meanwhile, 67.4% of the students believe that a lot now can be done to improve the lives of people with ADRD.

### 3.4. Relationship between Alzheimer's disease and dementia knowledge and attitude of medical students toward dementia

As shown in Table 6, the ADKS and DAS have a positive association (ρ = 0.227, p-value <0.001). A similar positive association between the overall ADKS and the scores of the dementia knowledge domain of the DAS was observed (ρ = 0.286, p-value <0.001). The scores of the assessment and diagnosis domain of the ADKS were positively associated with the overall DAS (ρ = 0.198, p-value <0.001), and the scores of the dementia knowledge domain of the DAS (ρ = 0.228, p-value <0.001). The scores of the caregiving domain of the ADKS were positively associated with the overall DAS (ρ = 0.204, p-value <0.001), the scores of the Social comfort domain of the DAS (ρ = 0.101, p-value = 0.046), and the scores of the dementia knowledge domain of the DAS (ρ = 0.229, p-value <0.001), respectively. On the other hand, the scores of the life impact domain of the DAS were negatively associated with the scores of the social comfort domain of the DAS (ρ = -0.113, p-value = 0.025). The scores of the course and treatment domains of the ADKS were also found to have a good association with the overall DAS and the scores of the dementia knowledge domain of the DAS.

Table 3. Number of Correct Answers of items on Alzheimer's disease knowledge scale (n = 393).

| ADKS Items | Correct Answer | n (%) | Pre-Clinical | Clinical | p-value[a] |
|---|---|---|---|---|---|
| **Symptoms:** | | | | | |
| 1. Tremor or shaking of the hands or arms is a common symptom in people with Alzheimer's disease. | False | 230 (58.5%) | 111 (51.6%) | 119 (66.9%) | 0.003* |
| 2. Trouble handling money or paying bills is a common early symptom of Alzheimer's disease. | True | 305 (77.6%) | 171 (79.5%) | 134 (75.3%) | 0.333 |
| 3. One symptom that can occur with Alzheimer's disease is believing that other people are stealing one's things. | True | 243 (61.8%) | 135 (62.8%) | 108 (60.7%) | 0.678 |
| 4. Most people with Alzheimer's disease remember recent events better than things that happened in the past. | False | 298 (75.8%) | 160 (74.4%) | 138 (77.5%) | 0.481 |
| **Risk factors:** | | | | | |
| 5. It has been scientifically proven that mental exercise can prevent a person from getting Alzheimer's disease. | False | 108 (27.5%) | 53 (24.6%) | 55 (30.9%) | 0.175 |
| 6. People in their 30s can have Alzheimer's disease. | True | 213 (54.2%) | 117 (54.4%) | 96 (53.9%) | 0.923 |
| 7. Having high cholesterol may increase a person's risk of developing Alzheimer's disease. | True | 220 (56%) | 122 (56.7%) | 98 (55%) | 0.760 |
| 8. Prescription drugs that help delay the progression of Alzheimer's disease are available. | False | 107 (27.2%) | 60 (27.9%) | 47 (26.4%) | 0.820 |
| 9. Having high blood pressure may increase a person's risk of developing Alzheimer's disease. | True | 193 (49.1%) | 108 (50.2%) | 85 (47.8%) | 0.685 |
| 10. Genes can only partially account for the development of Alzheimer's disease. | True | 238 (60.6%) | 124 (57.7%) | 114 (64%) | 0.214 |
| **Assessment and diagnosis:** | | | | | |
| 11. When a person with Alzheimer's disease becomes agitated, a medical assessment might reveal other health problems that caused the agitation. | True | 340 (86.5%) | 183 (85.1%) | 157 (88.2%) | 0.379 |
| 12. If trouble with memory and confusion starts suddenly, it is likely due to Alzheimer's disease. | False | 245 (62.3%) | 117 (54.4%) | 128 (71.9%) | <0.001* |
| 13. Symptoms of severe depression can be mistaken for symptoms of Alzheimer's disease. | True | 298 (75.8%) | 151 (70.2%) | 147 (82.6%) | 0.005* |
| **14. Alzheimer's disease is one type of dementia.** | True | 345 (**87.8%**) | 180 (83.7%) | 165 (92.7%) | 0.008* |
| **Course:** | | | | | |
| 15. After symptoms of Alzheimer's disease appear, the average life expectancy is 6 to 12 years. | True | 226 (57.5%) | 124 (57.7%) | 102 (57.3%) | 0.941 |
| 16. In rare cases, people have recovered from Alzheimer's disease. | False | 247 (62.8%) | 133 (61.8%) | 114 (64%) | 0.676 |
| 17. A person with Alzheimer's disease becomes increasingly likely to fall down as the disease gets worse. | True | 331 (84.2%) | 180 (83.7%) | 151 (84.8%) | 0.783 |
| 18. Eventually, a person with Alzheimer's disease will need 24-hour supervision. | True | 303 (77.1%) | 157 (73%) | 146 (82%) | 0.040* |
| **Life Impact:** | | | | | |
| 19. People with Alzheimer's disease are particularly prone to depression. | True | 341 (86.8%) | 179 (83.2%) | 162 (91%) | 0.025* |
| 20. Most people with Alzheimer's disease live in nursing homes. | False | 153 (38.9%) | 81 (37.7%) | 72 (40.0%) | 0.604 |
| 21. It is safe for people with Alzheimer's disease to drive, as long as they have a companion in the car at all times. | False | 305 (77.6%) | 165 (76.7%) | 140 (78.6%) | 0.716 |
| **Caregiving:** | | | | | |
| 22. People with Alzheimer's disease do best with simple instructions given one step at a time. | True | 310 (78.9%) | 165 (76.7%) | 145 (81.5%) | 0.266 |
| 23. When people with Alzheimer's disease begin to have difficulty taking care of themselves, caregivers should take over right away. | False | 71 (18%) | 36 (16.7%) | 35 (19.7%) | 0.511 |

(*Continued*)

**Table 3.** (Continued)

| ADKS Items | Correct Answer | n (%) | Pre-Clinical | Clinical | p-value[a] |
|---|---|---|---|---|---|
| 24. If a person with Alzheimer's disease becomes alert and agitated at night, a good strategy is to try to make sure that the person gets plenty of physical activity during the day. | True | 271 (68.9%) | 151 (70.2%) | 120 (67.4%) | 0.585 |
| 25. When people with Alzheimer's disease repeat the same question or story several times, it is helpful to remind them that they are repeating themselves. | False | 262 (66.7%) | 136 (63.2%) | 126 (70.8%) | 0.132 |
| 26. Once people have Alzheimer's disease, they are no longer capable of making informed decisions about their own care. | False | 118 (30%) | 58 (27%) | 60 (33.7%) | 0.152 |
| Treatment: | | | | | |
| 27. People whose Alzheimer's disease is not yet severe can benefit from psychotherapy for depression and anxiety. | True | 298 (75.8%) | 160 (74.4%) | 138 (77.5%) | 0.481 |
| 28. Poor nutrition can make the symptoms of Alzheimer's disease worse. | True | 338 (86%) | 185 (86%) | 153 (86%) | 0.979 |
| 29. When a person has Alzheimer's disease, using reminder notes can contribute to decline. | False | 152 (38.7%) | 76 (35.3%) | 76 (42.7%) | 0.146 |
| 30. Alzheimer's disease cannot be cured. | True | 323 (82.2%) | 172 (80%) | 151 (84.8%) | 0.235 |

\* Significant at P value <0.05.

[a] Chi-square test.

### 3.5. Factors affecting the medical students' knowledge and attitude toward dementia

Table 7 demonstrates the factors affecting the knowledge and attitude scores of the medical students in Palestine. Medical students in the clinical phase had higher ADKS scores than students in the pre-clinical phase (U = 14828, Z = 3.861, and p-value < 0.001). Medical students living in cities had higher DAS scores than students living in villages and refugee camps (p-value = 0.044). On the other hand, there was no significant association between the age groups, gender, marital status, experience with dementia, source of information, and the Alzheimer's disease Knowledge Score and the Dementia Attitude Scale scores.

## 4. Discussion

This study aimed to assess the knowledge of Alzheimer's disease and related dementia among medical students across Palestine, as well as to analyze their attitudes toward Dementia. Furthermore, we sought to compare the results between the pre-clinical and clinical students. Generally, our study's findings demonstrate limited knowledge of Alzheimer's disease and related Dementia among the medical students in Palestine, with a mean ADKS of 18.91 representing 63% of the correct answers. This finding was similar to the findings reported in previous studies in Jordan [16], Western India [17], and China [18,19]. Another study conducted in

**Table 4. Scores of content domain of Dementia Attitude Scale towards dementia (n = 393).**

| Domain | Range of total Scores | Mean ± SD | Mean correct rate (%) | Pre-Clinical, Mean Ranks | Clinical, Mean Ranks | U | Z | p-value[a] |
|---|---|---|---|---|---|---|---|---|
| Social Comfort Score | 10–70 | 44.27 ±7.61 | 63.2% | 198.46 | 195.24 | 18822 | 0.280 | 0.780 |
| Knowledge Score | 10–70 | 47.41 ±13.04 | 67.7% | 198.14 | 195.62 | 18889.5 | 0.219 | 0.827 |
| Total DAS | 20–140 | 91.68 ±16.06 | 65.5% | 196.68 | 197.39 | 19065 | 0.062 | 0.951 |

\* Significant at P value <0.05.

[a] Mann-Whitney U-test.

**Table 5. Mean and Number of responses to items on the Dementia Attitude Scale (n = 393).**

| Items | Disagree | Neutral | Agree | Mean ± SD |
|---|---|---|---|---|
| 1. It is rewarding to work with people who have ADRD. | 143 (36.4%) | 93 (23.7%) | 157 (39.9%) | 4.08 ±1.89 |
| 2. I am afraid of people with ADRD.a | 320 (81.4%) | 38 (9.7%) | 35 (8.9%) | 5.69 ±1.39 |
| 3. People with ADRD can be creative. | 172 (43.8%) | 84 (21.3%) | 137 (34.9%) | 3.89 ±1.72 |
| 4. I feel confident around people with ADRD | 165 (42%) | 129 (32.8%) | 99 (25.2%) | 3.76 ±1.55 |
| 5. I am comfortable touching people with ADRD. | 119 (30.3%) | 81 (20.6%) | 193 (49.1%) | 4.48 ±1.83 |
| 6. I feel uncomfortable being around people with ADRD.a | 275 (70%) | 54 (13.7%) | 64 (16.3%) | 5.24 ±1.59 |
| 7. Every person with ADRD has different needs. | 68 (17.3%) | 44 (11.2%) | 281 (71.5%) | 5.21 ±1.75 |
| 8. I am not very familiar with ADRD.a | 159 (40.5%) | 72 (18.3%) | 162 (41.2%) | 3.99 ±1.78 |
| 9. I would avoid an agitated person with ADRD.a | 197 (50.1%) | 71 (18.1%) | 125 (31.8%) | 4.48 ±1.70 |
| 10. People with ADRD like having familiar things nearby. | 94 (23.9%) | 89 (22.7%) | 210 (53.4%) | 4.59 ±1.74 |
| 11. It is important to know the past history of people with ADRD. | 76 (19.3%) | 35 (8.9%) | 282 (71.8%) | 5.31 ±1.87 |
| 12. It is possible to enjoy interacting with people with ADRD. | 86 (21.9%) | 61 (15.5%) | 246 (62.6%) | 4.84 ±1.78 |
| 13. I feel relaxed around people with ADRD. | 127 (32.3%) | 123 (31.3%) | 143 (36.4%) | 4.06 ±1.48 |
| 14. People with ADRD can enjoy life. | 102 (26%) | 61 (15.5%) | 230 (58.5%) | 4.65 ±1.71 |
| 15. People with ADRD can feel when others are kind to them. | 81 (20.6%) | 50 (12.7%) | 262 (66.7%) | 5.09 ±1.81 |
| 16. I feel frustrated because I do not know how to help people with ADRD.a | 128 (32.6%) | 79 (20.1%) | 186 (47.3%) | 3.82 ±1.81 |
| 17. I cannot imagine taking care of someone with ADRD.a | 219 (55.7%) | 78 (19.9%) | 96 (24.4%) | 4.66 ±1.63 |
| 18. I admire the coping skills of people with ADRD. | 108 (27.5%) | 108 (27.5%) | 177 (45%) | 4.36 ±1.69 |
| 19. We can do a lot now to improve the lives of people with ADRD. | 76 (19.3%) | 52 (13.3%) | 265 (67.4%) | 5.06 ±1.75 |
| 20. Difficult behaviors may be a form of communication for people with ADRD. | 106 (27%) | 83 (21.1%) | 204 (51.9%) | 4.40 ±1.66 |

Dementia Knowledge Score (Items: 3,7,10,11,12,14,15,18,19,20).

Social Comfort Score (Items:1,2,4,5,6,8,9,13,16,17).

a Reversely scored item.

Hong Kong using a different tool, the Alzheimer's Disease Knowledge (ADK) test, demonstrated poor knowledge of Alzheimer's disease and related dementia among medical students [20]. Conversely, the level of Alzheimer's disease knowledge of Palestinian medical students was lower than the level of knowledge of Australian medical students [21], Norwegian medical students [22], and medical students in the United Kingdom [23].

**Table 6. Relationship between dementia knowledge and attitude of medical students.**

| Alzheimer Disease Knowledge Scale | Dementia Attitude Scale | | | | | |
|---|---|---|---|---|---|---|
| | Overall score | | Social Comfort Score | | Dementia Knowledge Score | |
| | ρ | p-value[a] | ρ | p-value[a] | ρ | p-value[a] |
| Overall score | **0.227** | **<0.001*** | 0.002 | 0.971 | **0.286** | **<0.001*** |
| Symptoms | 0.057 | 0.261 | -0.045 | 0.377 | 0.079 | 0.117 |
| Risk Factors | 0.049 | 0.336 | 0.065 | 0.198 | 0.025 | 0.622 |
| Assessment & Diagnosis | **0.198** | **<0.001*** | 0.012 | 0.815 | **0.228** | **<0.001*** |
| Course | **0.139** | **0.006*** | - 0.065 | 0.198 | **0.205** | **<0.001*** |
| Life Impact | -0.020 | 0.687 | **-0.113** | **0.025*** | 0.020 | 0.700 |
| Caregiving | **0.204** | **<0.001*** | **0.101** | **0.046*** | **0.229** | **<0.001*** |
| Treatment | **0.140** | **0.005*** | 0.016 | 0.750 | **0.178** | **<0.001*** |

* Significant at P value <0.05.

[a] Spearman correlation by ranks.

**Table 7. Relationship between socio-demographic characteristics, knowledge, and Attitude of medical students towards people with dementia (n = 393).**

| Item | ADKS, Mean rank | U | Z | p-value | DAS, Mean rank | U | Z | p-value |
|---|---|---|---|---|---|---|---|---|
| **Age:** | | | | | | | | |
| Age < 20 | 177.31 | 9602 | 1.565 | 0.118[a] | 211.82 | 9928 | 1.173 | 0.241[a] |
| Age ≥ 20 | 201.5 | | | | 193.95 | | | |
| **Gender:** | | | | | | | | |
| Male | 185.51 | 17113 | 1.880 | 0.06[a] | 188.69 | 17695 | 1.353 | 0.176[a] |
| Female | 207.01 | | | | 204.24 | | | |
| **Medical Student Status:** | | | | | | | | |
| Pre-Clinical: | 176.97 | 14828 | 3.861 | < 0.001[*a] | 196.68 | 19065 | 0.062 | 0.951[a] |
| Clinical | 221.19 | | | | 197.39 | | | |
| **Marital status:** | | | | | | | | |
| Single | 196.16 | 834.5 | 1.188 | 0.235[a] | 198.05 | 756.5 | 1.465 | 0.143[a] |
| Married/Engaged | 251.42 | | | | 129.58 | | | |
| **Experience with dementia:** | | | | | | | | |
| Yes | 187.75 | 15.775.5 | 1.128 | 0.259[a] | 192.34 | 16364 | 0.565 | 0.572[a] |
| No | 201.47 | | | | 199.25 | | | |
| **Residence type:** | | | | | | | | |
| City | 202.97 | - | - | 0.135[β] | 207.66 | - | - | 0.044[*β] |
| Village | 182.58 | | | | 177.38 | | | |
| Camp | 223.26 | | | | 204.29 | | | |
| **Source of information:** | | | | | | | | |
| Internet | 185.07 | - | - | 0.136[β] | 195.83 | - | - | 0.210[β] |
| Curriculum | 208.43 | | | | 203.42 | | | |
| Awareness Campaigns | 203.00 | | | | 162.28 | | | |

* Significant at P value <0.05.

[a] Mann-Whitney U-test.

[β] Kruskal-Wallis test.

Our study revealed a significant difference in the ADKS scores between the pre-clinical and clinical students in Palestine, with the clinical students scoring higher ADKS scores than the Pre-Clinical students. This may be explained by the higher frequency of clinical students' exposure to dementia-related courses than the pre-clinical students. The study conducted by Wang et al demonstrated that Chinese medical students' knowledge of dementia was significantly associated with clinical practice and dementia education or training [18]. No significant differences in ADKS scores between males and females were found in our study, suggesting that gender does not significantly affect medical students' knowledge of dementia. This finding was similar to the findings reported in the study conducted by Gong et al among the undergraduate health professional students in China [19]. On the other hand, Wang et al reported that gender was significantly associated with Chinese medical students' knowledge of dementia [18]. The students' reported experience with Dementia turned out to be not a significant factor that affected their knowledge of dementia. Similar results were reported among Chinese Undergraduate Health Professional students [19]. We also found that marital status, type of residency, and source of information were not significantly associated with the medical students' knowledge of dementia.

Regarding the domains of the ADKS, the study's findings showed that the medical students in Palestine have a defect in the knowledge concerning the risk factors of developing Alzheimer's disease, with a mean correct rate of 45.7%, and in the knowledge concerning the

caregiving of patients with dementia, with a mean correct rate of 52.6%. This finding was similar to those reported among medical students in Western India [17]. On the other hand, the students' knowledge of the risk factors of dementia (2.74 (45.7%) ±1.28) and caregiving of patients with dementia (2.63 (52.6%) ±1.05) were lower than the reported level of knowledge concerning these domains in Norway (4.19 (69.8%) ±1.21, and 3.96 (79.2%) ± 1.01, respectively) and China (4.09 (68%) ± 1.06, and 3.08 (62%) ± 1.08, respectively) [19,22]. This may be attributed to the fact that about 67.4% of the Palestinian medical students who took part in our study had no experience with dementia. Our participants correctly answered 78% of the questions in the assessment and diagnosis domain, 70.7% of the questions in the treatment and management domain, and 70.5% of the questions in the course of the disease domain. This indicates that the medical students in Palestine have better knowledge of the diagnosis and management of Alzheimer's disease and this finding was supported by the findings reported from the study of the knowledge of Alzheimer's Disease among undergraduate healthcare students in Norwegian and Hong Kong [20,22]. However, the observed disparities in knowledge among medical students in Palestine, specifically their high level in diagnosis and management as opposed to their relative deficit in understanding risk factors, may be attributed to several factors. It could be due to the fact that the medical curriculum in Palestine places a greater emphasis on clinical aspects, diagnostic procedures, and therapeutic intervention, potentially enhancing students' competencies in these areas. Conversely, the emphasis on risk factors might be not integrated as extensively into the educational curriculum, resulting in a relative knowledge gap in this particular domain. Additionally, variations in available educational resources, faculty expertise, and the overall structure of medical education programs may contribute to divergent levels of knowledge acquisition across different facets of Alzheimer's disease. Clinical students tend to have higher scores in the diagnosis and management domain than pre-clinical students. Clinical students were more likely to correctly answer some of the questions in the other domains than the pre-clinical students; yet, they tend to have similar scores in the rest of the symptoms, risk factors, course, life impact, caregiving, and treatment domains of the ADKS.

Medical students in Palestine generally displayed a low level of positive attitude toward dementia with a mean DAS of 91.68 (±3.32 SD) out of 140 (65.5%). This finding was in line with the findings reported in previous studies of the attitude of the pre-clinical medical students at Columbia University (pre-interventional) [24], Chinese medical students [19], Australian medical students [21], and medical students in the United Kingdom (pre-interventional) [23], while medical students in Western India demonstrated relatively negative attitudes toward dementia [17]. Furthermore, the mean scores of Palestinian medical students in the Social Comfort domain of the DAS were slightly higher than the baseline mean of the Social Comfort domain for the pre-clinical students at Columbia University [24] and lower than the baseline mean for the medical students in the United Kingdom [23]. The mean scores of the dementia knowledge domain of the DAS for the Palestinian medical students were lower than the baseline mean for the pre-clinical students at Columbia University and the medical students in the United Kingdom [23,24]. Surprisingly, we found no significant differences in the scores of the DAS, the social comfort domain, and the dementia knowledge domain between the pre-clinical and clinical students in Palestine. Moreover, gender, marital status, experience with dementia, and source of information were not associated with the attitudes of the Palestinian medical students toward dementia. It was worth mentioning that the Palestinian medical students living in urban areas had higher DAS scores than those living in rural areas and refugee camps.

A positive correlation between the ADKS and DAS was found in our study ($\rho = 0.227$, p-value <0.001). A similar positive correlation between the ADKS and DAS was found among undergraduate healthcare students in China [19]. While our study found a positive association

between the ADKS and the scores of the dementia knowledge domain of the DAS ($\rho = 0.286$, p-value $<0.001$), no significant associations between the ADKS and the subdomains of the DAS were found in China [19]. This indicates that the higher level of dementia knowledge possibly affects the positive attitude of Palestinian medical students toward dementia. Notably, the scores of the Palestinian medical students in the "Assessment and Diagnosis", "Course", "Caregiving", and "Treatment" domains of the ADKS were positively correlated with the DAS and the dementia knowledge domain. In general, we highly recommend the evaluation of the current dementia curriculum in the medical faculties in Palestine and the incorporation of further training modules to build the capacities of the medical students in Palestine to ensure their empowerment to identify and properly manage cases of dementia.

### 4.1 Strengths and limitations

This is the first study that evaluates the Palestinian medical students' knowledge of dementia and their attitudes toward dementia. The participants in this study were diverse in terms of sociodemographic variables, such as gender, phase of medical training, and type of residency, making them more likely to represent the study population. The exclusion of first-year medical students from the study enabled us to assess the current level of knowledge and attitudes of the medical students based on the current medical curriculum. Our study has some limitations. First, our study was a cross-sectional observational study; we did not introduce any intervention in the form of an educational course or training program to improve the medical students' knowledge and attitudes toward dementia. Therefore, further prospective studies that include measures to assess the effectiveness of the medical education program on the improvement of the students' knowledge and correct their attitudes are needed. We recommend using the findings of this study as a reference to develop further interventional studies or training modules. Secondly, we collected the data using an online Google Form as a self-administered questionnaire; no interviews with the participants were done. This might have led to a social desirability bias, in which the participants answered the attitude-related questions not based on their true beliefs, but rather according to what they considered socially acceptable. Thirdly, we used validated tools to assess the knowledge of Alzheimer's disease and related dementia among medical students, but the ADKS contained 30 true or false questions, rather than case scenarios or multiple choice questions, which made it difficult to assess the real students' knowledge of dementia. Fourth, the questionnaire contained over 60 items across the three sections, which probably contributed to a lower response rate to the questionnaire.

## 5. Conclusion

Medical students in Palestine have limited knowledge of Alzheimer's disease and related Dementia and a relatively low level of positive attitude toward dementia. The higher the level of students' knowledge of Alzheimer's disease and related Dementia the more likely they tend to have a positive attitude toward dementia. There is a need for evaluation of the current dementia curriculum in the medical schools in Palestine and a need for the inclusion of more dementia training modules, especially in the pre-clinical phase. Further interventional studies to assess the impact of dementia Training/Education on the knowledge and attitudes of medical students in Palestine are needed.

### Supporting information

**S1 Data.**
(XLS)

## Acknowledgments

The authors would like to thank An-Najah National University (www.najah.edu) for the technical support provided to publish the present manuscript.

## Author Contributions

**Conceptualization:** Mohammad Abuawad, Ahmad Rjoub, Yazan Dumaidi, Motaz Daraghma, Mustafa Ghanim, Maha Rabayaa, Johnny Amer.

**Data curation:** Mohammad Abuawad, Ahmad Rjoub, Yazan Dumaidi, Motaz Daraghma, Mustafa Ghanim, Maha Rabayaa, Johnny Amer.

**Formal analysis:** Mohammad Abuawad, Ahmad Rjoub, Yazan Dumaidi, Motaz Daraghma, Mustafa Ghanim, Maha Rabayaa, Johnny Amer.

**Investigation:** Mohammad Abuawad, Ahmad Rjoub, Yazan Dumaidi, Motaz Daraghma, Mustafa Ghanim, Maha Rabayaa, Johnny Amer.

**Methodology:** Mohammad Abuawad, Ahmad Rjoub, Yazan Dumaidi, Motaz Daraghma, Mustafa Ghanim, Maha Rabayaa, Johnny Amer.

**Project administration:** Mohammad Abuawad, Ahmad Rjoub, Yazan Dumaidi, Motaz Daraghma, Mustafa Ghanim, Maha Rabayaa, Johnny Amer.

**Software:** Mohammad Abuawad, Ahmad Rjoub, Yazan Dumaidi, Motaz Daraghma, Mustafa Ghanim, Maha Rabayaa, Johnny Amer.

**Supervision:** Mohammad Abuawad, Ahmad Rjoub, Yazan Dumaidi, Motaz Daraghma, Mustafa Ghanim, Maha Rabayaa, Johnny Amer.

**Validation:** Mohammad Abuawad, Ahmad Rjoub, Yazan Dumaidi, Motaz Daraghma, Mustafa Ghanim, Maha Rabayaa, Johnny Amer.

**Visualization:** Mohammad Abuawad, Ahmad Rjoub, Yazan Dumaidi, Motaz Daraghma, Mustafa Ghanim, Maha Rabayaa, Johnny Amer.

**Writing – original draft:** Mohammad Abuawad, Ahmad Rjoub, Yazan Dumaidi, Motaz Daraghma, Mustafa Ghanim, Maha Rabayaa, Johnny Amer.

**Writing – review & editing:** Mohammad Abuawad, Ahmad Rjoub, Yazan Dumaidi, Motaz Daraghma, Mustafa Ghanim, Maha Rabayaa, Johnny Amer.

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
