## [Decision Letter · Decision Letter 0]

27 Feb 2024

PONE-D-24-03453Evaluation of Knowledge and Attitudes Regarding Alzheimer’s disease and Related Dementia among Medical Students in Palestine: A Cross-Sectional StudyPLOS ONE

Dear Dr. abuawad,

Thank you for submitting your manuscript to PLOS ONE. After careful consideration, we feel that it has merit but does not fully meet PLOS ONE’s publication criteria as it currently stands. Therefore, we invite you to submit a revised version of the manuscript that addresses the points raised during the review process.

We look forward to receiving your revised manuscript.

Kind regards,

Shahabedin Rahmatizadeh, Ph.D.

Academic Editor

PLOS ONE

Additional Editor Comments:

Considerable effort has been invested in conducting this research, which is commendable. However, the use of an English questionnaire in an Arabic-speaking country is an ambiguous issue. The reviewers suggest that it would have been better to translate the questionnaire into Arabic and evaluate it before use. Please address this issue and resubmit the revised article. Additionally, please address the other points mentioned by the reviewers.

Reviewers' comments:

Reviewer's Responses to Questions

**Comments to the Author**

1. Is the manuscript technically sound, and do the data support the conclusions?

Reviewer #1: Yes

Reviewer #2: Partly

Reviewer #3: Partly

Reviewer #4: Partly

Reviewer #5: Partly

2. Has the statistical analysis been performed appropriately and rigorously? 

Reviewer #1: Yes

Reviewer #2: Yes

Reviewer #3: Yes

Reviewer #4: Yes

Reviewer #5: Yes

3. Have the authors made all data underlying the findings in their manuscript fully available?

Reviewer #1: No

Reviewer #2: Yes

Reviewer #3: Yes

Reviewer #4: No

Reviewer #5: Yes

4. Is the manuscript presented in an intelligible fashion and written in standard English?

Reviewer #1: Yes

Reviewer #2: Yes

Reviewer #3: Yes

Reviewer #4: Yes

Reviewer #5: Yes

5. Review Comments to the Author

Reviewer #1: A good effort has been made to conduct the research, and the stages of the research have been well explained, but considering that they have used pre-approved questionnaires, it is not suitable for this journal.

It would have been better if the questionnaires were converted into the language and culture of the country where the study was conducted and its validity and reliability were re-evaluated.

And the following points were mentioned in the text that

Have the participants been able to have a proper understanding of the questions?

Are the questions consistent with the culture of your country?

Reviewer #2: The authors have provided a relatively clear and interesting manuscript, yet regrettably, it is poorly written. The objective of this research is to assess the understanding and perspective on Alzheimer's disease and related Dementia among medical students in Palestine. However, there are a couple of remarks that the authors should consider.

1- In the abstract of the article, the method section should also specify the sample size, the year of the study, and the analysis method.

2- In the abstract of the article, the result part is poorly written. Written conclusions are not drawn from the results.

3- Explain how the reliability of questionnaires is assessed in your community.

4- What formula is used to determine the sample size?

5- The sampling technique is incorrect and does not match the description of the Setting and Study Design. This method is employed when the units being studied are rare or not readily identifiable.

6- The names of the mentioned faculties are more than 5.

7- It should be specified in the tables which test was used.

8- In the discussion section, the values obtained from other studies should be mentioned. For example, in line 269 "level of knowledge concerning these domains in Norway and China", the average knowledge of the risk factors for each country should be mentioned.

9- In line 313: "The number of participants in this study exceeded the required sample size, which allowed us to generalize the data over the Palestinian medical students" Initially, the total population of your community needs to be determined, followed by determining the necessary sample size. while in the study, the mentioned items are unclear and this statement is incorrect.

10- Numerous findings have not been incorporated into the discussion; hence, the less significant results should either be excluded or included in the discussion.

11- References ought to be as current as feasible and should adhere to the citation style of the respective journal.

Reviewer #3: The aim of this article was to evaluate knowledge and attitudes about dementia and Alzheimer's disease among medical students. The authors have a relatively good analysis, but it needs to be revised.

The abstract of the article is not well written.

Line 68: In this section, the authors mentioned "The study took place among the Palestinian Medical Students enrolled in the “Medical Doctor” program in the five medical schools in Palestine", but in the following, 7 cases are mentioned.

Line 86: Do the authors have a justification for using the snowball sampling technique? Because using this sampling method is not suitable for this study.

Line 87 & 88: The method of calculating the sample size is ambiguous and there is no mention of the calculated size.

Line 127-130: The p-values related to the normality test should be mentioned.

Line 130: It is better to use Kruskal-Wallis test instead of the Kruskal-Wallis analysis of variance by ranks.

Line 136: The significance level of 5% is sufficient, and there is no need for the things stated in the rest of the sentence.

Table7: In Table 7, it is better to distinguish between the results of the Mann-Whitney and Kruskal-Wallis tests. Also, p-value related to the comparison of Pre-Clinical and Clinical should be corrected.

Not all reported results are discussed.

Reviewer #4: Manuscript revision

Title: Evaluation of Knowledge and Attitudes Regarding Alzheimer’s Disease and Related Dementia among Medical Students in Palestine: A Cross-Sectional Study.

1. Summary

A cross-sectional study was conducted among Palestinian medical students. Assessing knowledge and attitudes about dementia. Dementia Attitudes Scale (DAS) and Alzheimer's Disease Knowledge Scale (ADKS). As measured by the knowledge of dementia by the students, the overall mean score was ADKS was 18.91 out of 30. The mean ADKS score for students' dementia knowledge was 18.91 out of 30. Students' attitude towards dementia was measured by the DAS, with a mean score of 91.68 out of 140. Higher ADKS scores were observed in clinical students compared to pre-clinical students.

2. Comments

A good effort has been put into writing this manuscript. The manuscript provides valuable evidence regarding emphasize the necessity for assessing and upgrading the dementia curriculum in Palestinian medical schools, including adding training modules. However, to make the manuscript better, I have some comments as follows:

1. In the first line of the introduction section of the abstract, no need to There are no statistics. Importance. Dementia can be expressed in another way.

2. Considering that in the title written "Evaluation of Knowledge and Attitudes Regarding Alzheimer's disease and Related Dementia among Medical Students" Why are the clinical and pre-clinical phases compared in the results and discussion section?

3. In the "Setting and Study Design" section, 5 universities are written, but the names of 7 universities are written.

4. In line 120, it is written that an English questionnaire was used, Palestinian students are not native speakers and it may be difficult for them to understand the questions. How do you explain this?

5. This conclusion "Medical students in the clinical phase had higher ADKS scores than students in the pre-clinical phase" is clear. Why was it necessary to conduct a study in this field? Because due to the presence of medical students in the clinical phase, they go to internships and hospitals and have contact with patients, so their experience and knowledge It is more than phase students in pre-clinical.

6. Findings should not be repeated in the conclusion section. The conclusion section should be corrected.

Reviewer #5: The study evaluates knowledge and attitudes regarding Alzheimer's disease and dementia among medical students in Palestine using the Alzheimer’s Disease Knowledge Scale (ADKS) and Dementia Attitudes Scale (DAS). The study suggests the need for evaluation of the current dementia curriculum in medical schools and the inclusion of more dementia training modules to improve knowledge and attitudes among all students. Since the findings of this study can be a basis for other future studies, I consider it necessary to pay attention to the following recommendations, which are mentioned in the order of importance for the authors.

1. In this study, the snowball sampling method was used. The snowball sampling method is suitable for research where it is difficult for the researcher to identify people related to the subject. In the present study, it is not difficult to identify medical students to evaluate knowledge and attitudes about Alzheimer's disease and related dementia. Please explain why the authors used this sampling method.

2. It is mentioned in strengths and limitations: "The number of participants in this study exceeded the required sample size, which permitted us to generalize the data over the Palestinian medical students." It is important to mention that the snowball sampling method is one of the types of non-probability sampling. In non-probability sampling, the sample does not represent the community because more or less samples may be selected from some sections based on the researcher's opinion. Therefore, it seems that the mentioned sentence cannot be scientifically correct.

3. In section 2.1, states: "The study was conducted among Palestinian medical students enrolled in the "Doctor of Medicine" program at five Palestinian medical schools:", but the seven schools are named in the following sentence.

1. An-Najah National University,

2. Al-Azhar University-Gaza,

3. Palestine Polytechnic University,

4. Arab American University,

5. Islamic University of Gaza,

6. Al-Quds University,

7. Hebron University

Which one is correct? Five or seven?

4. The authors should include Alzheimer's disease, in addition to dementia, at the end of the introduction on page 4, line 64, as stated in the title.

5. In the introduction, line 37, additional parentheses are used to mention the source 1,2.

6. In Tables 4 and 6, the letter "l" is left in the term "Social Comfort Score".

6. PLOS authors have the option to publish the peer review history of their article (what does this mean?). If published, this will include your full peer review and any attached files.

Reviewer #1: No

Reviewer #2: No

Reviewer #3: No

Reviewer #4: No

Reviewer #5: No

---

## [Author Response · Author response to Decision Letter 0]

19 Mar 2024

Dear editor and reviewers, 

Thank you for your valuable comments. The manuscript has been edited, revised according to your comments. All comments have been addressed in the Response to reviewer's file.

BR

---

## [Decision Letter · Decision Letter 1]

15 Apr 2024

PONE-D-24-03453R1Evaluation of Knowledge and Attitudes Regarding Alzheimer’s Disease and Related Dementia among Medical Students in Palestine: A Cross-Sectional StudyPLOS ONE

Dear Dr. abuawad,

Thank you for submitting your manuscript to PLOS ONE. After careful consideration, we feel that it has merit but does not fully meet PLOS ONE’s publication criteria as it currently stands. Therefore, we invite you to submit a revised version of the manuscript that addresses the points raised during the review process.

We look forward to receiving your revised manuscript.

Kind regards,

Shahabedin Rahmatizadeh, Ph.D.

Academic Editor

PLOS ONE

Journal Requirements:

**Additional Editor Comments:**

Thank you for taking into consideration the issues and points raised by the article reviewers. It has been brought to our attention that the reviewers have raised several additional points that require addressing.

Reviewers' comments:

Reviewer's Responses to Questions

**Comments to the Author**

1. If the authors have adequately addressed your comments raised in a previous round of review and you feel that this manuscript is now acceptable for publication, you may indicate that here to bypass the “Comments to the Author” section, enter your conflict of interest statement in the “Confidential to Editor” section, and submit your "Accept" recommendation.

Reviewer #1: All comments have been addressed

Reviewer #2: All comments have been addressed

Reviewer #3: All comments have been addressed

Reviewer #5: (No Response)

2. Is the manuscript technically sound, and do the data support the conclusions?

Reviewer #1: Yes

Reviewer #2: Yes

Reviewer #3: Yes

Reviewer #5: (No Response)

3. Has the statistical analysis been performed appropriately and rigorously? 

Reviewer #1: Yes

Reviewer #2: (No Response)

Reviewer #3: Yes

Reviewer #5: (No Response)

4. Have the authors made all data underlying the findings in their manuscript fully available?

Reviewer #1: Yes

Reviewer #2: Yes

Reviewer #3: Yes

Reviewer #5: (No Response)

5. Is the manuscript presented in an intelligible fashion and written in standard English?

Reviewer #1: Yes

Reviewer #2: Yes

Reviewer #3: Yes

Reviewer #5: (No Response)

6. Review Comments to the Author

Reviewer #1: According to the changes made in the article, this article can be suitable with the corrections made.

Reviewer #2: Dear Author

Every comment has been addressed accurately. I possess just a single comment.

the abstract of the article, the method section should corrected:

“the Mann-Whitney U-test and the Kruskal-Wallis test were used to compare the mean between the groups with a 5% significance level” Instead “the Mann-Whitney U test and the Kruskal-Wallis test were used to compare the means between the groups with significance determined at a P value of less than”

Reviewer #3: A good effort has been put into editing this manuscript. However, to make the manuscript better, I have some comments as follows:

1. Line 154: Considering that your goal is to compare the mean attitude scores in different groups, there is no need to check the normality of the age variable.

2. Table7: The letters a, b related to different tests should be mentioned above the pـvalues.

Reviewer #5: (No Response)

7. PLOS authors have the option to publish the peer review history of their article (what does this mean?). If published, this will include your full peer review and any attached files.

Reviewer #1: No

Reviewer #2: No

Reviewer #3: No

Reviewer #5: No

---

## [Author Response · Author response to Decision Letter 1]

17 Apr 2024

Dear editor, 

All authors comments have been addressed. 

BR

---

## [Decision Letter · Decision Letter 2]

6 May 2024

Evaluation of Knowledge and Attitudes Regarding Alzheimer’s Disease and Related Dementia among Medical Students in Palestine: A Cross-Sectional Study

PONE-D-24-03453R2

Dear Dr. Mohammad Abuawad,

We’re pleased to inform you that your manuscript has been judged scientifically suitable for publication and will be formally accepted for publication once it meets all outstanding technical requirements.

Kind regards,

Shahabedin Rahmatizadeh, Ph.D.

Academic Editor

PLOS ONE

Additional Editor Comments (optional):

Reviewers' comments:

Reviewer's Responses to Questions

**Comments to the Author**

1. If the authors have adequately addressed your comments raised in a previous round of review and you feel that this manuscript is now acceptable for publication, you may indicate that here to bypass the “Comments to the Author” section, enter your conflict of interest statement in the “Confidential to Editor” section, and submit your "Accept" recommendation.

Reviewer #2: All comments have been addressed

Reviewer #3: All comments have been addressed

2. Is the manuscript technically sound, and do the data support the conclusions?

Reviewer #2: Yes

Reviewer #3: Yes

3. Has the statistical analysis been performed appropriately and rigorously? 

Reviewer #2: Yes

Reviewer #3: Yes

4. Have the authors made all data underlying the findings in their manuscript fully available?

Reviewer #2: Yes

Reviewer #3: Yes

5. Is the manuscript presented in an intelligible fashion and written in standard English?

Reviewer #2: Yes

Reviewer #3: Yes

6. Review Comments to the Author

Reviewer #2: (No Response)

Reviewer #3: I do not have any additional feedback to provide. The previous feedback and comments have been satisfactorily incorporated into the manuscript.

7. PLOS authors have the option to publish the peer review history of their article (what does this mean?). If published, this will include your full peer review and any attached files.

Reviewer #2: No

Reviewer #3: No

---

## [Editor Report · Acceptance letter]

8 May 2024

PONE-D-24-03453R2 

PLOS ONE

Dear Dr. Abuawad, 

I'm pleased to inform you that your manuscript has been deemed suitable for publication in PLOS ONE. Congratulations! Your manuscript is now being handed over to our production team.

Kind regards, 

on behalf of

Dr. Shahabedin Rahmatizadeh 

Academic Editor

PLOS ONE